# Estimation of Functional Reserve in Patients with Hospital-Associated Deconditioning

**DOI:** 10.3390/ijerph192114140

**Published:** 2022-10-29

**Authors:** Minhee Kim, Nackhwan Kim, Yuho Jeon, Seung-Jong Kim

**Affiliations:** 1Biomedical Research Center, Korea University Ansan Hospital, 123, Jeokgeum-ro, Danwon-gu, Ansan-si 15355, Korea; 2Korea University Research Institute for Medical Bigdata Science, Korea University Anam Hospital, Seoul 02841, Korea; 3Department of Biomedical Engineering, College of Medicine, Korea University, Seoul 02841, Korea

**Keywords:** human activities, cardiovascular deconditioning, physical functional performance, work capacity evaluation, exercise

## Abstract

Background: This study aimed to analyze the applicability of sit-to-stand (STS) muscle power tests for evaluating functional reserve in patients with hospital-associated deconditioning (HAD). Methods: This study is a single group preliminary observational study. STS tests were performed in the early stages of comprehensive rehabilitation treatment, and the interval changes in the clinical indicators were assessed after four weeks of clinical observation. A STS capacity ratio was estimated by the time duration of five STS repetitions (5r-STS) and the maximum number of STS repetitions over 30 s (30s-STS); the activities were measured using a three-dimension motion capture system and force plate. Results: After 4 weeks of comprehensive rehabilitation, the 10 m gait speed (*p* = 0.004), hand grip power (*p* = 0.022), hip extensor power (*p* = 0.002), Berg balance scale (*p* < 0.001), and modified Barthel index (MBI) (*p* = 0.001), respectively, were significantly improved. The force plate-derived (FPD) 30s-STS power and the number of repeats in the FPD 30s-STS showed a positive correlation with improvements in the hand grip power (Spearman’s Rho = 0.477, *p* = 0.045), hip extensor power (Spearman’s Rho = 0.482, *p* = 0.043), and MAI (Spearman’s Rho = 0.481, *p* = 0.043), respectively. The STS capacity ratio was correlated with higher improvements in the 10 m gait speed (Spearman’s Rho = 0.503, *p* = 0.034), hip extensor power (Spearman’s Rho = 0.494, *p* = 0.037), and MBI (Spearman’s Rho = 0.595, *p* = 0.009). Despite individual variability in the differences between the FPD and estimated STS power, the results for the correlation between the STS capacity ratio and clinical outcomes were consistent. Conclusions: The STS capacity ratio showed a positive correlation with the clinical outcomes, including gait speed, and may reflect a part of the functional reserve excluding the individual variability of performance.

## 1. Introduction

The remarkable deficits in physical function that occur after acute hospitalization are often defined as hospital-associated deconditioning (HAD) [1,2]. It has been reported that 68% of the inpatients in acute medical environments are discharged with a functional status lower than the pre-hospital level [3]; in addition, a recent meta-analysis study reported the combined prevalence across the included studies was 30% [4]. In particular, long-term bed rest, relative inactivity, sleep disturbances, and nutritional deficiencies due to hospitalization are related to the occurrence of HAD in the elderly [5,6,7,8].

The functional evaluation of patients with HAD provides important information for setting goals for comprehensive rehabilitation [9]. Both the goals and interventions of rehabilitation are based on a functional evaluation using multidisciplinary methods. As expected, decreased activity without any deformity or disability is a common finding in patients with HAD. Based on the World Health Organization’s International Classification of Functioning, Disability, and Health (ICF) [10], capacity and performance can be accounted for as factors in the evaluation of activity; the concept that the functional reserve between these two factors reflects the status of HAD has also been proposed [11].

The sit-to-stand (STS) test is an assessment tool for mobility-related function, generally among the older population [12]. There are several options for assessing STS performance, depending on whether the timing is set. Typically, the quantities measured are the time taken for five repetitions of standing from a seated position (5r-STS), and the maximum number of repetitions performed within 30 s (30s-STS) [13,14]. STS tests can be carried out using a familiar movement and conducted in restricted spaces; furthermore, they show good test–retest and inter-rater reliability [15]. The STS test usually assesses performance that comprehensively reflects the dynamic balance while standing up, sensory integration, and coordination capability, including the muscular function of the lower extremities at that time. In addition, the test with a time setting has been proposed as a tool that can replace cardiopulmonary exercise tests, which assess functional capacity [16].

However, the 5 STS test has floor effects in older adults with moderate to severe functional limitations [14], and several studies reported that the muscle power test more strongly predicted the functional limitations and mortality of older adults than the muscle strength test [17,18]. Therefore, in order to evaluate the functional capacity of HAD, which generally results in more severe limitations in older adults, an approach that can measure muscle power during the STS test is more appropriate than the time-based or repetition-based STS test. In addition, it was reported that the STS power test showed a stronger relationship with the physical performance of older adults than the hand grip, repetition-based STS test, and leg extension power [19]; however, clinical evidence of the STS power test for HAD remains lacking.

Therefore, the purpose of this study is to identify whether the STS power test can measure the functional reserve of patients with HAD. We hypothesized that the 5r-STS, 30 s STS power, and a ratio of two values are positively correlated with the initial or interval changes in other clinical outcomes. Furthermore, we provide a basis for selecting an applicable test method, according to the clinical environment, by estimating the agreement between the force plate-derived (FPD) STS power and estimated STS power.

## 2. Materials and Methods

### 2.1. Design and Participants

A prospective, observational study was conducted between September 2021 and January 2022 at a single institution. We tried to follow the Strengthening the Reporting of Observational Studies in Epidemiology (STROBE) guidelines. This study protocol was approved by a university-affiliated ethics committee (approval number 2021AS0265), and written informed consent was obtained from all patients. All procedures were performed according to the Declaration of Helsinki.

The subjects for data collection were limited to cases diagnosed by rehabilitation specialists as HAD, after recovery from acute medical conditions in a single university hospital, and were enrolled according to the following inclusion criteria: stable vital signs, including normoxemia, for longer than 72 h without any supplies; grade 3 or higher muscle strength in both lower extremities; cognitive function within the normal range (mini-mental state exam score ≥ 25); and the Berg balance scale (BBS) score < 45 [20]. The exclusion criteria were as follows: neuropathological abnormalities; BBS score ≤ 20; evidence of orthostatic hypotension; moderate to severe dizziness; anatomical deformity or contracture of lower extremities; visual field defects; and consent denial.

The participants in this study received a total of 4 weeks of comprehensive rehabilitation treatment, which was individualized as a protocol suitable for the initial functional level and implemented five times a week, 40 min a day. The STS tests were performed only before the rehabilitation treatment. The patients performed the 5r-STS initially and the 30s-STS after a 24-h wash-out period.

The sample size of this study was calculated based on the software G power 3.1.7 (Franz Faul, Universit€at Kiel, Germany, 2007) [21]. The effect size was selected to be 0.65 for a medium effect size, according to Cohen’s recommendation (1988), with the power set at 80% and the type I error fixed at 5%. The results indicated that a minimum of 16 participants was needed. To allow for about 10% attrition, the sample size was increased to 18 participants.

### 2.2. Sit-to-Stand Test

In the starting position, the participant was seated in a standardized armless chair (0.47 m height), with their back straight and their arms crossed at the wrists and held against their chest. After the participant was informed of the requirements and precautions, the test began with the cue. They were instructed to stand and sit fully before standing as quickly as possible for a total of five repetitions, and the time taken to complete the five repetitions was recorded. After the wash-out period of 24 h, the participant was asked to repeat as many of the sit-to-stand actions as possible in 30 s, and the maximum number completed was recorded. The test protocols included a warm-up and cool-down exercise. The participants were given verbal encouragement during the task. The minimum value of three trials of the 5r-STS and the maximum value of two trials of the 30s-STS were considered the participant’s score.

### 2.3. STS Power and Capacity Ratio

To measure the FPD STS power, the time required and the number of repetitions were measured using a three-dimensional motion capture system with ten cameras, at a sample rate of 100 Hz (OptiTrack, NaturalPoint, Inc., Corvallis, OR, USA). To capture the STS concentric phase, which was divided into three phases (Figure 1), reflective markers were placed on the subject’s body, following the Helen Hayes marker set model, and two additional markers were placed on the right and left greater trochanters to define the time at which the buttock was off the chair [22]. The force values were obtained from force platforms with a sample rate of 1000 Hz (BMS400600-2000, AMTI, Watertown, MA, USA). The force (N) was estimated as the mean value of the vertical ground reaction forces (GRF) during the STS tasks. The velocity was calculated as the ratio between the vertically displaced distance of the center of body mass and the time duration of the STS concentric phase. The distance of the vertical displacement was directly measured as the difference from the floor to the greater trochanter in the chair-sitting and standing positions. The power of the STS movement was estimated using the previously validated equation [23]:$$\mathrm{FPD}\;\mathrm{STS}\;\mathrm{power}\;=\;\mathrm{GRF}\;\times \frac{\mathrm{Leg}\;\mathrm{length}-\mathrm{chair}\;\mathrm{height}\;}{\;\mathrm{concentric}\;\mathrm{time}}$$

The estimated STS power was calculated as the body mass (total body mass minus shank and feet mass, 0.9 body mass (kg) multiplied by *g* (9.81m/s^2^)) multiplied by the mean velocity. The estimated STS mean velocity (m/s) was calculated as the vertical distance between the leg length (0.5 body height) and the height of the chair, divided by the mean time required to complete the concentric phase of the one STS repetition displaced during the test [24].
$$5r\text{-}\mathrm{STS}\;\mathrm{power}=\frac{\mathrm{Body}\;\mathrm{mass}\;\times \;0.9\;\times \;g\;\times \;\left[\mathrm{Height}\;\times \;0.5-\mathrm{Chair}\;\mathrm{height}\right]}{\left(\frac{\mathrm{Total}\;\mathrm{STS}\;\mathrm{time}\;\mathrm{duration}}{5}\right)\times \;0.5}$$
$$30s\mathrm{STS}\;\mathrm{power}=\frac{\mathrm{Body}\;\mathrm{mass}\;\times \;0.9\;\times \;g\;\times \;[\mathrm{Height}\;\times \;0.5-\mathrm{Chair}\;\mathrm{height}]}{\left(\frac{30s}{\mathrm{Number}\;\mathrm{of}\;\mathrm{STS}\;\mathrm{repeats}}\right)\times \;0.5}$$

A ratio of the STS capacity was then obtained from the following equation:$$\mathrm{STS}\;\mathrm{capacity}\;\mathrm{ratio}=\frac{30s\text{-}\mathrm{STS}\;\mathrm{power}\;}{5r\text{-}\mathrm{STS}\;\mathrm{power}\;}$$

### 2.4. Clinical Outcomes

*Anthropometrics*. The skeletal muscle index (SMI) was measured using a bioimpedance analysis (BIA-101, Akern-RJL Systems, Florence, Italy) as the ratio of the appendicular skeletal muscle mass to the square of height (kg/m^2^).

*Maximal gait speed*. The maximal gait speed was measured at the maximal safe walking pace over a distance of 10 m while providing strong verbal encouragement [24]. The time required to complete the test was recorded in units of 0.1 s and converted into speed (m/s).

*Muscle strength*. The maximal hand grip strength was measured using a Jamar dynamometer (Sammons Preston Rolyan, Chicago, IL, USA). The subjects were seated in a chair without arm rests, with the elbow flexed at an angle of 90° and the forearm supported by a horizontal surface. Strong verbal encouragement was given during each test. Two trials of hand grip tests were conducted for each hand, and the highest recording from all the trials was used for the analysis. A hand-held dynamometer (Sammons Preston, IL, USA) was used to measure the hip extensor strength with maximum effort. The subjects were in the prone position with hips and knees extended, and the dynamometer was placed on the posterior aspect of the shank, near the ankle joint [24]. The subjects then maximally extended their lower body without bending the knees or trunk for 5 s. Two trials were performed with a 3-min rest period between the trials, and the highest value was used for the analysis.

*Mobility performance*. Dynamic balance was assessed using the timed up-and-go (TUG) test [25]. The time taken to complete the test was recorded.

*Gait balance*. The Berg balance scale (BBS) is a frequently used performance-based ordinal scale that assesses postural balance [26].

*Activities of daily living*. The modified Barthel index (MBI), which consists of 10 items with a maximum score of 100, was used to measure the activities of daily living [27].

### 2.5. Statistical Analysis

The Shapiro–Wilk test was used to determine the normality of the distribution of all the data. Standard descriptive statistics were used for the continuous variables. A paired *t*-test was used to compare the baselines and follow-up demographics and functional performance data, as well as to test the correlation between 5r-STS/30s-STS and the initial or interval change of functional performance. The correlation coefficient values between ±0.1 and ±0.3, ±0.4 and ±0.6, and >0.7 were considered weak, moderate, and strong, respectively [28]. As a secondary analysis, to assess the agreement between the FPD and estimated STS tests, Bland–Altman plotting was used, and the level of agreement (LOA) was calculated [24]. All the statistical analyses were performed using SPSS (version 22.0; SPSS Korea Data Solution Inc., Seoul, Korea). The statistical significance was set at *p* < 0.05.

## 3. Results

### 3.1. Patient Characteristics

Overall, 25 participants were evaluated to verify their eligibility for this study. Finally, 18 adults with a mean age of 65.3 years satisfied the inclusion/exclusion criteria. The reasons for ineligibility included a BBS score ≤ 20 (*n* = 4), moderate to severe dizziness (*n* = 2), and consent denial (*n* = 1). The clinical diagnosis in the acute phase of the 18 participants was as follows: 8 with pneumonia; 6 with an acute kidney injury; 2 with acute cholecystitis; 1 with a urinary tract infection; and 1 with cellulitis on the leg. The baseline and follow-up data have been reported in Table 1.

During the 4-week follow-up period, statistically significant improvements were noted in the 10 m gait speed, hand grip power, hip extender power, BBS, and MBI.

### 3.2. Outcomes

The results of the FPD 5r-STS and FPD 30s-STS tests are presented in Table 2. The mean concentric time of the time taken in the 5r-STS and 30s-STS and the mean GRF during the tasks were measured. The FPD STS power and capacity ratio were calculated.

In Table 1, the interval change (percent, %) of the statistically significant variables (10 m gait speed, hand grip power, hip extensor power, BBS, and MBI) was set as the independent variables, and the correlation between the initial variables of the physical function and the results of the FPD STS tests was analyzed. Higher values of the initial 10 m gait speed were correlated moderately with higher improvements in the hand grip power (Spearman’s Rho = 0.550, *p* = 0.018) (Figure 2A). Higher scores of the initial BBS were correlated moderately with higher improvement of hip extensor power (Spearman’s Rho = 0.600, *p* = 0.008) (Figure 2B). Higher points of the initial MBI were correlated moderately with higher improvement of hand grip power (Spearman’s Rho = 0.576, *p* = 0.012) and hip extensor power (Spearman’s Rho = 0.541, *p* = 0.021) (Figure 2C).

This study analyzed whether the time taken for the FPD 5r-STS and the number of repeats in the FPD 30s-STS, which are the conventional results in STS tests, correlated with the improved variables. In addition, the power of each test and the STS capacity ratio estimated in this study were included in the correlation analysis. Higher power in the FPD 30s-STS was correlated moderately with higher improvements in the hand grip power (Spearman’s Rho = 0.477, *p* = 0.045) (Figure 3A). Higher repeats in the FPD 30s-STS were moderately correlated with improved hip extensor power (Spearman’s Rho = 0.482, *p* = 0.043) and MBI (Spearman’s Rho = 0.481, *p* = 0.043) (Figure 3B). A higher-STS capacity ratio was correlated moderately with a higher improvement in the 10 m gait speed (Spearman’s Rho = 0.503, *p* = 0.034), hip extensor power (Spearman’s Rho = 0.494, *p* = 0.037), and MBI (Spearman’s Rho = 0.595, *p* = 0.009) (Figure 3C). The power and time taken in the 5r-STS were not statistically correlated with any improved variables.

Additionally, a Bland–Altman plot showing the agreement between the FPD STS power and the estimated STS power is presented in Figure 4. Specifically, the 5r-STS (bias: 23.65, upper LOA: 98.23, lower LOA: −50.93) power and 30s-STS power (bias: 45.50, upper LOA: 170.41, lower LOA: −79.5), and the STS capacity ratio (bias: 0.09, upper LOA: 0.43, lower LOA: −0.25), revealed some individual variability in the distance covered between them; however, only the 30s-STS power was scattered in range of LOA. 

## 4. Discussion

The purpose of this study was to analyze the applicability of STS muscle power tests for evaluating functional reserve in patients with HAD. To achieve this purpose, we investigated the correlation between the initial or interval changes in clinical outcomes related to functional performance and 5r-STS and 30s-STS power. Our main results showed that the STS capacity ratio, which means a gap between the 30s-STS and the 5r-STS, can reflect the functional reserve of patients with HAD. This equation can be a simple yet reliable indication of reduced functional reserves. Additionally, to evaluate an individual’s STS function, 30s-STS is more reproducible and relatively reliable as it is not significantly affected by the application of a force plate.

The reduced activity in patients with HAD is remarkably similar to the decrease in physical function experiencing geriatric frailty. Although geriatric frailty has been defined in numerous manners, there exists a consensus that the main characteristics of frailty are a decline in physical reserves and a collapse in homeostasis, which increases the risk of negative prognoses [29,30,31]. This concept is also proposed in the “Activity” domain of the ICF and may be interpreted to mean capacity and performance. Functional limitations and negative prognoses in patients with HAD result from a significant decrease in capacity that limits full performance; this cannot be explained using the domain “Function and Structure” alone. Therefore, the diagnostic features of HAD may include a reduced gap between capacity and performance. This study proposed a ratio capacity to overcome individual variabilities in capacity.

The reasons why STS was selected as an intervention for analyzing activity are as follows: an easy, daily, and repeatable behavior; the variables measured with less error by observation, and safe to perform. The ability to stand up from a chair is a basic function for maintaining functional independence [32]. The STS movement is inherently unstable because it changes rapidly from a stable seating position to a position with a relatively small support base and a high center of mass. Mechanical difficulties in the performance of STS are likely to pose significant challenges to groups with reduced muscle strength [33]. Also, the STS test has been shown to have good reliability in the elderly [23]. However, the results of the STS require a standardization process for a relative comparison because the individual variability is significant [34]. Our study defined ratio capacity using the average value of 5r-STS as a parameter, which is similar to the level of general STS performance in an individual.

The 30s-STS is a time-limited test of physical function and is associated with other measures of functional capacity, independence of daily living, and frailty [12,35]. This test has been proposed as an advantage in that it can be recorded, even if the STS is performed only once for those subjects who cannot perform the number-based STS test [14]. Interestingly, if STS is performed more than five times, the environmental conditions of a limited time seem to encourage the subject to exert maximum endurance [36]. This study focused on this point, assuming that the decrease in patients’ endurance in mild or moderate levels of deconditioning was reflected in the results of the 30s-STS and expected to reflect a part of functional reserve. The existing studies have suggested that endurance-trained exercise is an intervention therapy that restores functional reserve [11,37]. The STS capacity ratio may include functional reserve in consideration of the individual variability in STS performance. The estimate has a higher statistical correlation than the 5r-STS and 30s-STS with the functional prognosis in the other assessment tools. In particular, the correlation with the prognosis of gait speed is clinically significant. Gait speed has been identified as the “sixth vital sign” of geriatric assessment [38], and functionally independent older adults experienced hospitalization-associated declines in gait speed [39].

The main result of this preliminary study is that the gap between the 30s-STS and the 5r-STS can reflect the functional reserve of patients with HAD. Although it is difficult to correlate reduced functional reserves with the mean performance and capacity for the 5r-STS and 30s-STS, respectively, at least some of the functional reserve can be reflected by offsetting the differences in the STS performance among individuals. The ratio capacity is a standardized value of the difference in the power required for the same activities with different rules. This concept could offer a route for establishing the diagnostic nature of HAD.

In particular, the improved values for the hand grip power, hip extensor power, and MBI after 4 weeks of comprehensive rehabilitation are positively correlated with the FPD 30s-STS power but not with the FPD 5r-STS power. These results suggest that the FPD 30s STS power reflects the functional reserve under HAD to a greater extent than the 5r-STS power. In addition, despite individual variability in the differences between the FPD and estimated 30s-STS power, they show a higher agreement than the 5r-STS power. Thus, our results suggest that the 30s-STS power is a useful clinical evaluation tool for reflecting and predicting the functional activity under HAD; moreover, it can be measured easily using a simple formula, without expensive devices that require significant technical skills.

Among the limitations of this preliminary study, the small sample size, gender imbalance, and narrow range of functional levels of HAD are issues that need to be addressed in further studies. In addition, further large prospective cohort study that can classify ranges of STS muscle power values according to the level of functional ability of HAD is needed. Nevertheless, the present study has clinical significance as a preliminary study for determining whether the functional reserve in patients with HAD can be evaluated using STS tests.

## 5. Conclusions

This study was designed to analyze the availability of the STS test for evaluating the functional reserve of patients with HAD, by identifying the correlation with improvements in other clinical outcomes after 4 weeks of comprehensive rehabilitation. Further, the bias between the FPD and estimated STS power was calculated to provide the basis for selecting an applicable test method. The results of this study indicate that the STS capacity ratio can reflect the functional reserve of patients with HAD, which could offer a route to establishing the diagnostic characteristics of HAD.

## Figures and Tables

**Figure 1 ijerph-19-14140-f001:**
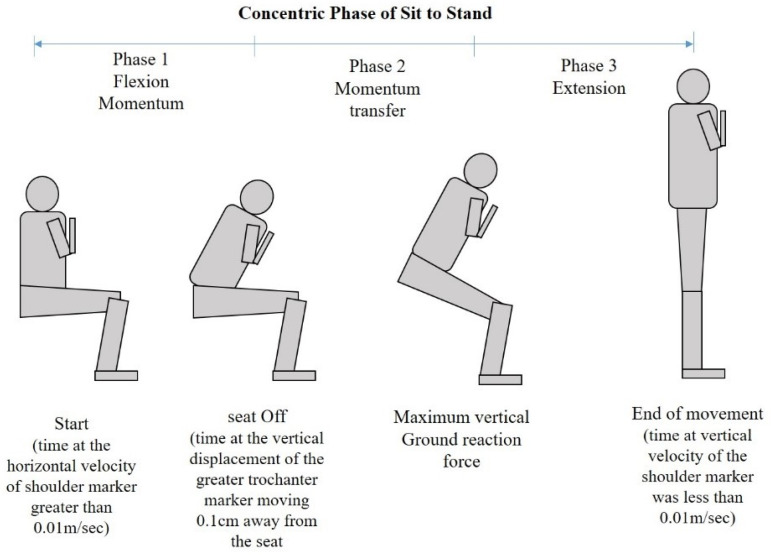
Concentric phase of sit to stand. Phase 1 is the flexion momentum, which began with the initiation point of the trunk and ended with seat-off. Phase 2 is the momentum transfer, which began with seat-off and ended when the maximum vertical ground reaction force occurred. Phase 3 is the extension phase, which began with the maximum vertical GRF and ended with the movement.

**Figure 2 ijerph-19-14140-f002:**
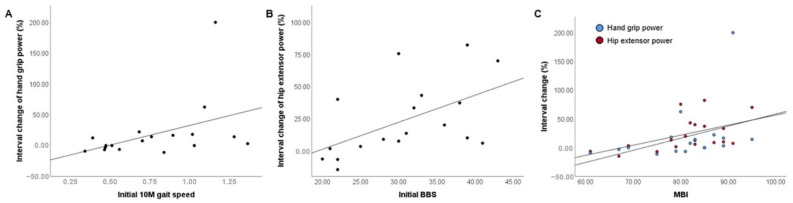
Scatter plots for initial values of physical assessment and significantly improved variables. (**A**) Scatter plot for the first 10 m gait speeds and hand grip power interval changes. (**B**) Scatter plot for initial BBS and hip extensor power interval change. (**C**) Scatter plot for initial MBI and hip extensor power interval change.

**Figure 3 ijerph-19-14140-f003:**
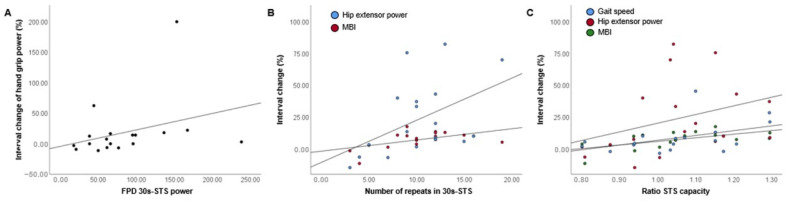
Scatter plots for FPD 30s-STS power, number of repeats in FPD-30s-STS, STS capacity, and significantly improved variables. (**A**) Scatter plot for FPD 30s-STS power and interval change of hand grip. (**B**) Scatter plot for the number of repeats in 30s-STS, hip extensor power interval change, and MBI. (**C**) Scatter plot for STS capacity ratio, gait speed interval change, hip extensor power, and MBI.

**Figure 4 ijerph-19-14140-f004:**
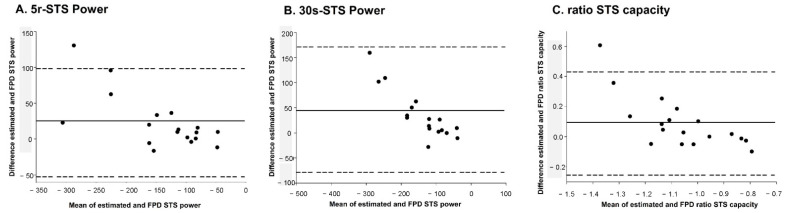
Bland–Altman plots showing differences between estimated and FPD 5r-STS power (**A**) and between estimated and FPD 30s-STS power (**B**) and between estimated and FPD STS capacity ratio (**C**). The solid line is the mean difference; the upper and lower dotted lines represent the upper and lower 95% limits of agreement, respectively.

**Table 1 ijerph-19-14140-t001:** Baseline and follow-up data for patients with HAD.

Variables	Before the Rehabilitation	After Four Weeks	*p*-Value
Age (years)	65.28 ± 7.45 range: 55 to 80)	not investigated	-
Sex (male:female)	12:6	not investigated	-
Height (cm)	164.59 ± 8.48	not investigated	-
Weight (kg)	61.42 ± 12.24	61.6 ± 11.8	0.727
BMI	22.59 ± 3.50		
MMSE	29.61 ± 0.78	not investigated	-
SMI (kg/m^2^)	8.71 ± 1.25	8.7 ± 1.2	0.458
10 m gait speed (m/s)	0.79 ± 0.32	0.84 ± 0.31	0.004 *
Hand grip (dominant, kg)	20.61 ± 10.02	22.2 ± 9.2	0.022 *
Hip extensor (maximal, kg)	5.41 ± 2.52	6.4 ± 2.7	0.002 *
TUG (s)	15.57 ± 6.82	14.9 ± 7.2	0.306
BBS	30.67 ± 7.51	40.4 ± 10.4	<0.001
MBI	81.05 ± 8.71	87.33 ± 12.85	0.001
Time duration of 5r-STS (s)	17.57 ± 8.47	not investigated	-
Repeats of 30s-STS (number)	10.33 ± 4.13	not investigated	-

* Significant difference (*p* < 0.05). MMSE, mini-mental state examination; SMI, skeletal muscle index; TUG, timed up and go; BBS, Berg balance scales; MBI, modified Barthel index.

**Table 2 ijerph-19-14140-t002:** Results of STS tests for patients with HAD.

Patients	FPD 5r-STS			FPD 30s-STS			STS Capacity Ratio
	Concentric Time	GRF (N)	Power (N·m/s)	Concentric Time	GRF (N)	Power (N·m/s)	
1	3.83	533.72	42.13	4.65	520.16	33.82	0.80
2	1.80	493.89	87.27	1.65	449.71	86.45	0.99
3	1.28	494.69	119.41	1.20	495.06	127.60	1.07
4	1.07	647.70	171.63	0.96	672.11	198.34	1.16
5	1.86	627.17	142.60	1.49	668.43	189.06	1.33
6	1.72	649.59	119.34	1.60	626.45	123.57	1.04
7	2.06	636.56	98.88	2.27	669.48	94.50	0.96
8	1.33	624.93	145.69	1.62	563.85	108.51	0.74
9	0.96	601.96	158.61	0.78	615.40	200.40	1.26
10	1.16	587.96	165.30	0.98	588.94	194.91	1.18
11	1.28	937.25	317.29	0.82	1010.08	532.01	1.68
12	1.42	881.76	273.61	1.25	894.66	315.68	1.15
13	1.10	924.02	352.61	1.00	883.87	369.97	1.05
14	3.16	489.39	51.95	3.60	490.30	45.66	0.88
15	2.64	742.76	84.19	3.13	729.70	69.66	0.83
16	1.61	576.95	89.37	1.13	607.25	133.99	1.50
17	1.63	948.25	256.50	1.51	1029.11	300.64	1.17
18	1.52	492.27	88.62	1.37	514.32	103.09	1.16

NOTE. Values are raw data of all participants. FPD, force plate-derived; STS, sit to stand; GRF, ground reaction forces.

## Data Availability

Not applicable.

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
