# Peer review of "Estimation of Functional Reserve in Patients with Hospital-Associated Deconditioning"

_ijerph, 2022, doi:10.3390/ijerph192114140_

Round 1
Reviewer 1 Report
The paper deals with a very interesting topic: the applicability of sit-to-stand (STS) muscle power tests for evaluating functional reserve in patients with hospital-associated deconditioning (HAD). As noted by the authors, long-term bed rest, relative inactivity, sleep disturbances, and nutritional deficiencies due to hospitalization are related to the occurrence of HAD in the elderly. These patients require appropriately selected rehabilitation procedures, adapted to their physical fitness. The functional evaluation of patients with HAD provides important information for setting goals for comprehensive rehabilitation. The tests proposed and analyzed by the authors are easy to carry out, repeatable, safe and do not burden patients with reduced physical fitness.
I believe that the article is well written, the authors chose the appropriate statistical methods and presented the obtained results in an understandable way.
Thank you for the opportunity to review this well constructed study. However, I have some suggestions for improving the manuscript:
1. In the introduction (line 34) there is no reference to the position of the literature; describing the definition of HAD, on what bibliographic position they are based;
2. Most of the literature contains quite old items (over 5-10 years) - please check if more recent literature is available and if so, supplement / change it;
3. As in the preliminary point above (line 34), the authors quote, "that 68% of the inpatients in acute medical environments are discharged with a functional status lower than the pre-hospital level" - these are data from 2009, do the authors have access to more recent data ?;
4. Regarding the subchapter Material and methods: please provide a more precise description / description of the research group. What acute diseases were the patients after? did they refer to diseases / disorders of the respiratory, cardiovascular and neurological systems?
5. Table 1 - please complete the information below the table to expand the abbreviations of the analyzed parameters (MMSE, SMI, TUG, BBS, MBI, 5r-STS, 30s-STS). This way the table will be easier to read;
6. Similarly, in Table 2-, please complete the information below the table to expand the abbreviations of the analyzed parameters (GRF, STS, FPD ...);
7. In Tables 1 and 2 some results are presented to one decimal place, and some to two decimal places. Please, standardize the presented results;
8. The manuscript lacks information on how many patients were initially enrolled in the study and how many were disqualified, and for what reason. How many patients did not meet the eligibility criteria (how many achieved MMSE <25; how many BBS ≤20, how many had dizziness, contractures, etc.);
9. What was the minimum and what was the maximum age of the respondents ?;
10. It would be worth adding BMI to the assessment;
11. Are there norms for each age group for the analyzed tests? and if so, how many and how many did the results within the normal range, and has the situation changed after rehabilitation?
Thank you
Author Response
Dear Reviewers,
The authors would like to thank the editor and anonymous reviewers for their efforts to their productive review and valuable comments. We tried our best to apply to the reviewer’s comments faithfully and make changes in yellow where necessary in manuscript file.
Please download the file attached.
Thanks again for your criticism.
Sincerely,
From Authors.

Reviewer 2 Report
Title: the title of study didn’t reflect the study design and method, need to restructure
Abstract: Mention the study design in the methodology part
Add the statistical results in the result section
The keywords should be according MeSh
Introduction
The literature review is weak, need to improve it
Need to explain the study rational, what are the gaps in the previous study and how this study going to cover those gaps
The study hypothesis and purpose not clear,
Methodology
Explain the dependent and independent variables
Why you have 4 weeks’ intervention? And only measure the pre-test
The effect size of sample size calculation need reference
Results:
In the first section of results show the difference between pre-test and posttest, which in not present in the methodology. In addition, maybe your study design be the quasi-experimental
The results were presented in this part not mention in the study purpose and hypothesis
Discussion
In the first section of the discussion need to explain your main finding with study objective.
Need to explain the strength and practical implication of study
Reference:
May this paper can help you in the present study
http://www.ijahm.com/index.php/IJAHM/article/view/23
Author Response

(The authors gave the same response as above.)

Reviewer 3 Report
The manuscript entitled “Estimation of functional reserve in patients with hospital-associated deconditioning” presents preliminary study conclusions on the applicability of sit-to-stand (STS) muscle power tests to evaluate functional reserve in hospitalized deconditioned patients (HAD). Patients with HAD experience functional limitations and poor prognosis as a result of a marked decline in capacity that prevents full performance. A reduce gap between capacity and performance may be considered one of the diagnostic indicators of HAD. Considering that 68% of hospitalized patients with acute conditions have a lower functional status at discharge than at admission, the examined topic is very important.
The study followed the Strengthening the Reporting of Observational Studies in Epidemiology (STROBE) guidelines, and every procedure was carried out according to the Declaration of Helsinki. The study protocol was approved by the ethics committee of Korea University Ansan Hospital, and all participants provided written informed consent.
A fundamental skill to preserve functional independence is the ability to stand up from a chair. People with reduced muscle strength are likely to face significant challenges due to mechanical issues during STS. In this study, STS tests were performed early in comprehensive rehabilitation treatment and changes in clinical indicators were evaluated after four weeks of clinical observation. The authors estimated a ratio capacity by the time duration of five STS repetitions (5r-STS), test performed initially, and the maximum number of STS repetitions over 30 seconds (30s-STS), test performed after a 24-hour wash-out period, and associated with measures of functional capacity, independence of daily living, and frailty.
To evaluate functional reserve in patients with HAD, the authors measured STS power, skeletal muscle index, 10-meter gait speed, maximum hand grip strength, hip extensor strength, mobility performance using the Time Up and Go test (TUG), gait balance using Berg Balance Scale (BBS) and ADLs using modified Barthel Index (MBI).
Analyzing data collected from 18 adults, the results showed a statistically significant improvement in five of the measured parameters: 10-meter gait speed, hand grip strength, hip extensor strength, BBS and MBI.
Gait speed is considered an important element of the geriatric evaluation. The authors found a clinically significant correlation of gate speed with prognosis. Statistically, positive correlations were identified between higher values of the initial gate speed and the improvement in hand grip power, and between the higher STS capacity ratio and the higher improvement in gait speed.
Positive correlations have also been identified between higher initial BBS scores and improvement of hip extensor power, higher initial MBI points, and higher improvement in hand grip and hip extensor power. In 30s-STS, higher power was correlated with higher improvements in hand grip power and higher repetition with improved hip extensor power. Higher-ratio STS capacity was correlated with hip extensor power and MBI.
A decrease in patient endurance in mild or moderate deconditioning was shown in the 30s-STS results and was expected to reflect a part of functional reserve.
The study demonstrated that the functional reserve of HAD patients can be reflected by ratio STS capacity, which could provide a path to determine the diagnostic characteristics of HAD. The 30s-STS power has proven to be a useful clinical evaluation tool to reflect and predict functional activity in HAD.
The article presents a current topic of general interest.
Bibliographic sources are cited appropriately.
The information is well organized, in a logical sequence, and demonstrates good documentation and analysis of the collected data. The conclusions are supported by the data presented.
I appreciate the article as interesting, documented, and elaborated with meticulousness and correctness and, therefore, I recommend its publication in its current form, without other changes.
Author Response

(The authors gave the same response as above.)

Round 2
Reviewer 2 Report
It can be publish